# Carbon Adsorbents Obtained from Pistachio Nut Shells Used as Potential Ingredients of Drinking Water Filters

**DOI:** 10.3390/molecules28114497

**Published:** 2023-06-01

**Authors:** Agata Wawrzyniak, Małgorzata Wiśniewska, Piotr Nowicki

**Affiliations:** 1Department of Radiochemistry and Environmental Chemistry, Faculty of Chemistry, Institute of Chemical Sciences, Maria Curie-Sklodowska University in Lublin, M. Curie-Sklodowska Sq. 3, 20-031 Lublin, Poland; agawaw7@st.amu.edu.pl; 2Department of Applied Chemistry, Faculty of Chemistry, Adam Mickiewicz University in Poznań, Uniwersytetu Poznańskiego 8, 61-614 Poznań, Poland

**Keywords:** pistachio nut shells, activated biocarbons, direct activation, chemical activation, adsorption, methylene blue, poly(acrylic acid)

## Abstract

Water resources are increasingly degraded due to the discharge of waste generated in municipal, industrial and agricultural areas. Therefore, the search for new materials enabling the effective treatment of drinking water and sewage is currently of great interest. This paper deals with the adsorption of organic and inorganic pollutants on the surface of carbonaceous adsorbents prepared by thermochemical conversion of common pistachio nut shells. The influence of the direct physical activation with CO_2_ and chemical activation with H_3_PO_4_ on parameters, such as elemental composition, textural parameters, acidic–basic character of the surface as well as electrokinetic properties of the prepared carbonaceous materials was checked. The suitability of the activated biocarbons prepared as the adsorbents of iodine, methylene blue and poly(acrylic acid) from the aqueous solutions was estimated. The sample obtained via chemical activation of the precursor turned out to be much more effective in terms of all the tested pollutants adsorption. Its maximum sorption capacity toward iodine was 1059 mg/g, whereas in relation to methylene blue and poly(acrylic acid) 183.1 mg/g and 207.9 mg/g was achieved, respectively. For both carbonaceous materials, a better fit to the experimental data was achieved with a Langmuir isotherm than a Freundlich one. It has also been shown that the efficiency of organic dye, and especially anionic polymer adsorption from aqueous solutions, is significantly affected by solution pH and temperature of the adsorbate–adsorbent system.

## 1. Introduction

The presence of organic compounds in water results from various types of natural processes, precipitation, surface runoff, industrial and municipal wastewater [1,2,3,4]. The occurrence of these substances is undesirable for many reasons. Organic compounds give the liquid phase certain physical properties, such as turbidity, color, smell, taste or suspension formation. They can also disturb the biological balance, and this affects the course of self-purification of waters. The biggest problem is the organic compounds contained in the water, which are responsible for the formation of disinfection by-products. They react with chlorine and its derivatives, often creating toxic substances [5].

Due to the high biological harmfulness of sewage, both municipal and industrial, it must be treated in sewage treatment plants before being discharged into a water reservoir [6]. Industrial wastewater is generated during the technological processes of many types of industries. In order to assess their toxicity, it is necessary to know not only the composition of their general outflow from the entire industrial plant, but also the composition of streams flowing out of individual production departments. Knowledge of the quantitative and qualitative parameters of wastewater allows to estimate the necessary degree of purification, at which discharge to a natural receiver will not deteriorate the water purity class. The problem of sewage is particularly acute in food industry, petrochemical plants, tanneries, pulp mills, dairies and sugar factories, as they pose a great threat to natural receivers [7,8,9,10,11,12,13]. The most common organic components of wastewater include proteins, carbohydrates, fats, oils, resins, dyes, phenols, petroleum products, detergents, pesticides, and synthetic polymers. Foaming of sewage is caused by the presence of surface-active substances (washing agents, complexing agents, bleaching agents, inhibitors, stabilizers, and optical brighteners), which reduce the surface tension of water. The group of cationic surfactants, which are highly toxic, is the most difficult to degrade using a biological method [14].

In wastewater treatment processes, mechanical, chemical, biological, mixed methods and disinfection are used [15]. Depending on the type of wastewater, the treatment process should be planned in such a way that it achieves the highest possible degree of purification at minimum cost. A typical wastewater treatment process generally consists of four stages: mechanical, biological, removal of biogenic compounds and the so-called water renewal. Despite the use of multi-stage treatment, some non-degradable impurities may remain in the wastewater. These substances are usually referred to as refractive compounds. These are organic micropollutants of sewage that are biochemically difficult to decompose or are not degradable. Of the organic compounds, chlorine, nitro, amide and cyano derivatives of aromatic and aliphatic compounds are particularly not susceptible to biological decomposition [16,17].

Removal of impurities from water, including refractive ones, can be successfully carried out using adsorption on the activated carbon surface [18,19,20,21]. The constant increase in the amount and variety of hazardous substances in wastewater makes the standard sorbents less and less effective. A highly developed specific surface area, appropriate porosity and a variety of surface groups of activated carbon make it a versatile and effective adsorbent of organic and inorganic compounds of various structure and molecule size. In addition to excellent adsorption properties, the production of activated carbon is relatively cheap, especially when bio-precursors, such as sawdust, nut shells, fruit stones or waste parts of plants, e.g., herbs, are used for its preparation [22,23,24,25].

Industrial production of activated carbon usually involves two-step physical (also called thermal) activation or one-step chemical activation of precursors, such as wood, peat, lignite or bituminous coal [26,27]. However, due to ecological reasons, waste biomass is increasingly used for the production of activated carbon, due to which it does not have to be burned or stored in landfills. One of the plant-derived materials used for the large-scale production of activated carbon are coconut shells [28,29,30]. However, analysis of the literature data shows that pistachio nutshells also seem to be an interesting precursor for its production. Carbon materials obtained as a result of pistachio nutshells pyrolysis, physical activation with the use of CO_2_ or steam and chemical activation with the use of ZnCl_2_, K_2_CO_3_, KOH, NaOH, NH_4_NO_3_, H_2_SO_4_ or HCl have turned out to be effective adsorbents of various types of organic and inorganic pollutants. Activated carbons obtained in this way can be successfully used for the removal of NO_2_ and H_2_S [31], SO_2_ [32], VOCs [33], organic dyes [34], phenolic compounds [35], parabens [36], pharmaceuticals [37], cyanide [38], heavy metal ions [39], transition metal ions [40], etc. In addition, they can be used as promising electrode material for electrochemical capacitors [41] and catalysts or catalyst supports [42].

However, in the literature, there are no reports on the application of activated carbons obtained from pistachio shells as adsorbents of polymeric compounds that are commonly found in municipal and industrial wastewater. Taking the above into account, the main objective of the present study was to produce a series of new activated biocarbons via direct physical and chemical activation of common pistachio nut shells as well as to assess their usefulness for the removal of inorganic and organic pollutions (dyes and polymers) from the aqueous solutions. The influence of thermochemical conversion of biomass on parameters, such as elemental composition, type of porous structure as well as acidic–basic character of the surface were also investigated.

## 2. Results and Discussion

### 2.1. Elemental Composition of the Starting Pistachio Nut Shells as well as Activated Biocarbons Prepared via Direct Physical and Chemical Activation

Analysis of the data presented in Table 1 shows that both physical and chemical activation of pistachio nut shells significantly changed the percentage share of carbon and other elements in relation to the precursor. The content of elemental carbon as a result of thermochemical treatment increased almost twice, with a simultaneous significant decrease in the content of hydrogen and oxygen. In case of nitrogen and sulfur, smaller differences were observed. PNSAP (product of direct physical activation) and PNSAc (product of chemical activation) samples differ quite significantly from each other, which is related to a different mechanism of reaction between biomass and CO_2_ or H_3_PO_4_. As a result of the thermochemical conversion of biomass, the content of mineral admixtures in the carbonaceous structure increased significantly; interestingly, it increased much more in case of the carbon obtained via chemical activation with orthophosphoric acid.

### 2.2. Acidic–Basic Properties of the Precursor and the Activated Biocarbons Prepared

In order to fully characterize the chemical character of the carbonaceous materials surface, the content of functional groups of basic and acidic nature was determined and the pH value of their aqueous extract was measured. According to the data collected in Table 2, the precursor and both activated biocarbons differ significantly in terms of acidic–basic properties. The total content of surface functional groups varies in the range 0.77–1.01 mmol/g, while the pH value ranges from 3.14 to 9.11. The precursor used shows slightly acidic character (pH = 5.29) and has three times greater number of acidic than basic groups.

Each of the obtained activated biocarbons contains groups of both acidic and basic nature, however, their mutual ratio changes dramatically depending on the procedure of activation. Sample PNSAc obtained via chemical activation of pistachio nut shells contains above 13 times more acidic groups than basic ones. In turn, the sample PNSAp obtained by means of direct physical activation of the precursor is characterized by a clear predominance of basic groups. Such a significant difference is a consequence of the diverse thermal treatment conditions applied during the activation process as well as the chemical nature of the activating agents used. Carbon dioxide in combination with the high activation temperature (800 °C) is conducive to the generation of basic groups, whereas orthophosphoric acid used as an activating agent at a significantly lower temperature (typically 500–600 °C) promotes the formation of acidic surface species.

### 2.3. Surface and Electrokinetic Properties of the Activated Biocarbons Prepared

The activated biocarbons’ surface charge density (σ_0_) and zeta potential (ζ) changes with the increasing solution pH are presented in Figure 1. The analysis of σ_0_ dependencies (Figure 1a) allows determination of points of zero charges (pH_pzc_), that is, the specific pH values at which the surface charge assumes value zero. For PNSAp adsorbent, the pH_pzc_ is 10.8, whereas for PNSAc, it is 4.4. Such results demonstrated the acidic character of the PNSAc sample obtained via chemical activation of pistachio nut shells and the basic properties of the physically activated PNSAp material.

The curves of zeta potential changes as a function of solution pH have a completely different course (Figure 1b) in comparison to the solid surface charge dependencies. The isoelectric points (pH_iep_) are located at pH values 4.8 for the PNSAp activated carbon and 3.2 for the PNSAc carbonaceous material. These pH_iep_ values differ from the pH_pzc_ values obtained for analogous systems (especially for physically activated PNSAp sample). The main reasons for the different compositions of stiff surface and movable diffusion parts of electrical double layers (edl) formed inside activated carbon pores are the overlapping of edls created on parallel pore walls and the presence of impurities in the structure of adsorbents [43].

### 2.4. Textural and Morphological Parameters of the Carbonaceous Materials Obtained from Pistachio Nut Shells

According to the data summarized in Table 3 and in Figure 2, both carbon adsorbents obtained by thermochemical conversion of pistachio nut shells show completely diverse textural parameters. Physical activation of the precursor turned out to be not a very effective solution in terms of porous structure development. The surface area of the PNSAp sample is only 31 m^2^/g, and the total pore volume is 0.059 cm^3^/g. The course of the pore size distribution curve (Figure 2) shows that the porous structure of the PNSAp biocarbon consists of small amounts of mesopores and macropores. Such unfavorable textural parameters may indicate that the activation conditions were too mild for this precursor. Most likely, a longer activation time, a higher processing temperature or a reduction in the size of the precursor grains should be used. In turn, chemical activation of pistachio nut shells with H_3_PO_4_ turned out to be a very effective procedure for porous structure generation. BET surface area of the PNSAc sample significantly exceeds 1200 m^2^/g, and the total pore volume reaches almost 1.4 cm^3^/g. In contrast to the product of physical activation, the sample activated with orthophosphoric acid has a significant amount of micropores and small mesopores in its structure, which promises well in terms of its application for adsorption purposes.

The SEM images presented in Figure 3 confirm the textural and morphological differences between the activated biocarbons prepared via direct physical and chemical activation of pistachio nut shells. Both samples differ significantly in terms of the number, size, shape and the arrangement of holes and slits. Surface of the PNSAp sample is practically smooth and contains only a few and completely randomly arranged pores. On the other hand, in case of the PNSAc sample, a well-developed porous system can be seen. The brighter fragments observed for both samples may be due to the presence of mineral admixtures (ash) or other activation by-products.

### 2.5. Sorption Performance of the Activated Biocarbons Prepared from Pistachio Nut Shells in Relation to Iodine and Methylene Blue

In order to check the usefulness of the obtained carbonaceous materials for removal of impurities with a small molecule size from the aqueous environment, the iodine number was determined. The data collected in Figure 4 illustrate a significant effect of the method of activation on the sorption abilities towards iodine. Sample PNSAc obtained as a result of chemical activation has definitely greater potential for removing small molecular impurities (diameter close to 2 nm) from the liquid phase than the product of direct physical activation, which is most probably a consequence of its better developed porous structure.

It should also be emphasized that the result obtained with the PNSAc sample is comparable or even better than for many commercial products, for which the iodine number usually ranges from 800 to 1000 mg/g.

The second variant of testing the adsorption properties of the activated biocarbons obtained from pistachio nut shells was the assessment of their ability to remove organic dyes from aqueous solutions, based on the removal of methylene blue, which is a synthetic thiazine dye of cationic character. The results of the sorption tests are presented in Table 4. According to these data, the sample obtained as a result of chemical activation of the precursor, which was able to adsorb 183.11 mg of methylene blue per gram of adsorbent, turned out to be the more effective adsorbent. The product of direct physical activation was able to adsorb only 9.04 mg of methylene blue, which is of course related to its poorly developed porous structure.

From the further analysis of the data collected in Table 4, it can be seen that the Langmuir model isotherm fits the experimental data more accurately than the Freundlich one, as indicated by closer to unity values of the correlation coefficient R^2^ (especially in case of the PNSAc sample). Thus, it can be assumed that methylene blue adsorption is most probably realized via monolayer coverage of the adsorbent surface by the organic dye molecules present in the aqueous environment, in particular for the PNSAp sample. In case of the material activated with H_3_PO_4_, the adsorption mechanism seems to be more complicated, because a fairly high value of R^2^ was also obtained for the Freundlich model, which assumes the formation of a multilayer of adsorbate molecules on the adsorbent surface.

According to the data presented in Figure 5, one of the parameters that influence the efficiency of methylene blue removal from aqueous solutions is pH of the solution. In case of the PNSAp sample, the increase in pH results in an improvement of the adsorption capacity in the entire investigated range, reaching a maximum at pH equal to 10. For the sample obtained via chemical activation of pistachio nut shells, a slightly different trend of changes is observed. Increase in solution pH from 4 to 6 results in quite significant decrease in the MB removal efficiency, whereas further change of pH (in the range 6–10) leads to an increase in the efficiency of dye removal from the liquid phase.

The efficiency of methylene blue adsorption also depends on the temperature of the adsorbent–adsorbate system (Figure 6). However, the influence of this parameter is less pronounced than for the pH of the solution. In case of the PNSAp sample, the adsorption capacity achieved at temperature of 20 °C and 40 °C differs only by 0.7 mg/g (so it is within the margin of error). For sample activated with H_3_PO_4,_ an increase in the temperature of the system to 30 °C results in an slight improvement in the sorption capacity of the PNAc sample by 3.5 mg/g, however, a further increase in temperature to 40 °C brings the opposite effect.

### 2.6. Sorption Properties of the Activated Biocarbons Prepared towards the Poly(acrylic acid) Polymer

Figure 7a presents the adsorption isotherms of poly(acrylic acid) on the examined activated carbons surface obtained at pH 6 and at 25 °C. The PAA adsorbed amounts are significantly greater in case of the PNSAc carbonaceous material. It should be noted that at pH 6, the polymeric carboxyl groups show practically complete dissociation and under such pH conditions they repeal with the PNSAc surface and attract with the PNSAp surface. Despite unfavorable electrostatic conditions for PAA adsorption in the case of PNSAc sample, the polymer amount adsorbed is considerably higher than that of PNSAp activated carbon. This proves that besides electrostatic forces, the chemical interactions can be responsible for polymer binding with the solid surface groups. In such a system, hydrogen bonds can also be formed [44]. The total content of surface groups are higher in the case of chemically activated material (Table 2), which additionally favors more effective binding of macromolecules with the activated carbon surface. Moreover, totally dissociated PAA chains at pH 6 assume a more developed conformation and their adsorption on the positively charged PNSAp surface lead to the flat structures formation. It then leads to the blockade of the activated carbon surface groups making them inaccessible for other polymeric macromolecules (small adsorption is observed). In turn, in the case of negatively charged PNSAc surface, the structure of PAA adsorption layer is characterized by the larger packing of macromolecules adsorbed perpendicularly to the solid surface, which is manifested in higher adsorption level.

The PAA adsorption kinetics curves obtained at pH 6 and at 25 °C are presented in Figure 7b. In the case of both examined activated biocarbons, the adsorption equilibrium is reached after 6 h. It is quite long time, but in the case of polymeric chains such behavior is typically observed. Large polymer macromolecules undergo the process of reconformation after the initial phase of bonding with the surface, and after a certain time they reach their equilibrium conformation.

Adsorption of poly(acrylic acid) with slight acidic character depends considerably on the solution pH. As can be seen in Figure 8a, the PAA amounts adsorbed on both activated carbons decrease with the increasing pH (in the range 2–12). The maximal adsorption was obtained at pH 12 and it is 207.9 mg/g for PNSAc and 61.2 mg/g for the PNSAp sample. The previous studies indicated that the hydrodynamic radius of PAA chains at pH 3 is 0.77 nm [45] (them assume coiled conformation in the solution due to the negligible dissociation of PAA carboxyl groups) and thus polymeric coils can effectively penetrate the activated carbon pores (the large adsorption is noticed). With the increasing solution pH, the conformation of PAA macromolecules becomes more and more developed, which considerably limits the possibility of their entry into the carbonaceous material pores.

The conformational changes of polymeric macromolecules can also occur as a result of temperature changes. In the case of poly(acrylic acid), the hydrodynamic diameter of its chains in the solution at pH 6 changes from 2.7 nm at 25 °C to 3.1 nm at 35 °C [46]. This is the main reason behind the decrease in PAA adsorption observed with the increasing temperature in the range 25–35 °C (Figure 8b). The increase in linear dimensions of poly(acrylic acid) molecules limits their adsorption inside the pores, and as such it can only occur on the outer surface of activated carbon particles.

Based on the data presented in Table 5, it can be concluded that activated biocarbons obtained via direct physical and especially chemical activation of pistachio nutshells perform well in terms of adsorption of organic and inorganic pollutants from the aqueous solutions. Particularly noteworthy is the fact that the adsorption capacities of the PNSAc and PNSAp samples towards poly(acrylic acid) exceed most of the results reported so far in the literature. The product of chemical activation fares much better in this respect, therefore, further research should focus on optimizing its production procedure.

## 3. Materials and Methods

### 3.1. Activated Biocarbons Preparation

The pistachio nut shells (PNS, Figure 9a) used as the starting material in this work came from stores located in the Wielkopolska region (Poland). In the first step, the precursor was washed with distilled water and then air-dried and crushed to a size of 5 mm. Next, the fragmented starting material was subjected to two different activation procedures: (1) direct physical activation with carbon dioxide including simultaneous pyrolysis and activation stage); and (2) chemical activation with orthophosphoric acid.

Physical activation (Ap) was conducted in the horizontal laboratory furnace equipped with a quartz tubular reactor (Czylok, Jastrzębie-Zdrój, Poland). Approximately 10 g of the crushed pistachio nut shells were placed in the nickel boat and subjected to thermal treatment under CO_2_ atmosphere. The sample was placed in the furnace preheated to a temperature of 800 °C and annealed for a period of 30 min. Carbon dioxide (technical CO_2_ 2.8, Linde Gaz Polska, Kraków, Poland) was used as the activating agent (flow rate 250 cm^3^/min). After the designated activation time had elapsed, the boat was pulled from the hot zone of the furnace and cooled down to room temperature in an inert-gas atmosphere (technical nitrogen 4.0, flow rate 170 cm^3^/min, Linde Gaz Poland, Kraków, Poland). The activated biocarbon sample was next washed with boiling distilled water and dried to constant weight at 110 °C. The obtained carbonaceous material was denoted as PNSAp (Figure 9b).

Chemical activation (Ac) was carried out according to the following procedure: in the beginning the crushed pistachio nut shells were mixed with 50% solution of orthophosphoric acid (Avantor Performance Materials, Gliwice, Poland) at the precursor–activator weight ratio equal to 2:1. After the impregnation stage (24 h at room temperature, with occasional stirring) the sample was dried at 110 °C in order to evaporate water. Next, the impregnated precursor was placed into the quartz boat and heated in the horizontal laboratory furnace in the nitrogen atmosphere (flow rate 330 cm^3^/min). The activation procedure consisted of 5 steps: (1) heating to 200 °C at the rate of 5 °C/min, (2) annealing of the sample at that temperature for 30 min, (3) heating to the final activation temperature of 500 °C, (4) annealing of the sample for 30 min and (5) cooling down to room temperature under the nitrogen flow. Finally, the sample was washed with hot distilled water and dried at 110 °C. The activated biocarbon obtained in this way was designated as PNSAc (Figure 9c).

### 3.2. Characterization of the Precursor and Activated Biocarbons

The elemental analysis of the starting material as well as products of its thermochemical conversion was performed using the CHNS Vario EL III instrument (Elementar Analysensysteme GmbH, Langenselbold, Germany) according to the procedure described in detail in our previous work [51].

The ash (mineral matter) content for all materials under investigation was determined according to the ISO 1171:2002 standard, using the microwave muffle furnace Phoenix (CEM Corporation, Matthews, IL, USA).

Characterization of the pore structure of the pistachio nut shells-based activated biocarbons was made by the nitrogen adsorption–desorption measured at −196 °C on the sorptometer Autosorb iQ (Quantachrome Instruments, Boynton Beach, FL, USA). Before the isotherm measurement, the sample was degassed under vacuum at a temperature of 300 °C for 12 h. Specific surface area (S_BET_) was evaluated in the range of relative pressure (p/p_0_) between 0.05 and 0.30, whereas total pore volume (V) was calculated by converting the amount of liquid N_2_ adsorbed at a relative pressure of 0.99. Average pore diameter (D) for the carbonaceous materials prepared was calculated from dependence D = 4V/S_BET_. Pore size distribution for each activated biocarbon was determined based on the BJH model. Moreover, the commonly known t-plot method was applied to determine micropore volume and micropore surface area.

The surface morphologies of the activated biocarbons were analyzed using the high-resolution electron microscope Quanta 250 FEG (FEI, Waltham, MA, USA).

The acidic–basic character of the starting pistachio nut shells and activated biocarbon’s surface was evaluated according to the Boehm titration method, described in detail in our previous paper [51]. Volumetric standards of 0.1 mol/dm^3^ hydrochloric acid or sodium hydroxide (Avantor Performance Materials, Gliwice, Poland) were used to neutralize groups of basic or acidic nature, respectively. The pH value of the precursor and both activated biocarbons aqueous extracts was determined using the CP-401 pH-meter (ELMETRON, Zabrze, Poland) equipped with an EPS-1 combination glass electrode, according to the procedure described in [51].

Additionally, the potentiometric titration method [60] was applied for the determination of the changes of surface charge density (σ_0_) of activated carbons as a function of solution pH. The obtained curves enable determination of pH_pzc_ (pzc—point of zero charge) of the carbonaceous materials. At pH value equaled to pH_pzc_, the total surface charge is zero, which means that the concentrations of positively and negatively charged surface groups are the same. For this purpose, 50 cm^3^ of the suspension containing 0.35 g of PNSAp and 0.016 g of PNSAc were prepared. The examined systems were titrated in the pH range 3–11 using NaOH with the concentration of 0.1 mol/dm^3^ and automatic Dosimat 765 micro-burette (Metrohm, Opacz-Kolonia, Poland). The changes in suspension pH after each portion of the added base were monitored using the glass and calomel electrodes (Beckman Instruments, Brea, CA, USA) and pH 240 pH-meter (Radiometer, Warsaw, Poland). The measurements were carried out at 25 °C (thermostated Teflon vessel, RE 204 thermostat, Lauda Scientific, Lauda-Königshofen, Germany). The changes in the σ_0_ value as a function of solution pH were calculated with the special computer program “Titr_v3”.

The zeta potential (ζ) of the activated carbons particles was determined using the Doppler laser electrophoresis method and the Zetasizer Nano ZS (Malvern Instruments, Malvern, UK). The measured electrophoretic mobility of the solid particles was related to the zeta potential (ζ) using Henry’s equation [61]. The curves showing the zeta potential changes as a function of solution pH enable the determination of the pH_iep_ (iep—isoelectric point) of the examined materials. At pH value equaled to pH_iep_, the total charge gathered in the slipping plane area assumes a value of zero, meaning that the concentrations of positively and negatively charged ions/groups in this region of the electrical double layer are equal. A 200 cm^3^ of the suspension containing 0.05 g of the activated carbons was prepared. This system was next subjected to the action of ultrasounds (XL 2020 ultrasonic head, Misonix, Farmingdale, NY, USA) for 3 min and divided into several parts. In each of them, the specific pH value (2, 3, 4, 6, 8, 10 or 12) was adjusted (Φ360 pH—meter, Beckman, Brea, CA, USA). The electrokinetic measurements were performed at 25 °C.

### 3.3. Adsorption of Iodine, Methylene Blue and Poly(acrylic acid)

Three adsorbates of completely different physicochemical properties were used in order to characterize the sorption abilities of the pistachio nut shells-based activated biocarbons. The model impurities included iodine aqueous solution (Avantor Performance Materials, Gliwice, Poland); methylene blue aqueous solution, a synthetic cationic dye representing organic aromatic compounds (Avantor Performance Materials, Gliwice, Poland); and poly(acrylic acid) aqueous solution representing anionic polymers (Fluka, St. Louis, MO, USA).

The iodine adsorption number of the activated biocarbons was determined according to the PN-83/C-97555.04 national standard according to the procedure described in detail in [51].

Adsorption of methylene blue (MB) from aqueous solution was performed using the procedure detailed in [62]. Effect of initial MB concentration, pH (changing in the range 4.0–10.0) and temperature of the solution (20, 30 and 40 °C) on the efficiency of adsorption of the above-mentioned organic dye from the liquid phase were determined. The dye concentration in the solution after adsorption tests were established using a double beam UV–Vis spectrophotometer Cary 100 Bio (Agilent, Santa Clara, CA, USA) at the wavelength of 664 nm, using the previously prepared calibration curve. Distilled water was applied as a reference sample.

The adsorbed amount of methylene blue (*q_ads_*, mg/g) was calculated according to the Equation (1):(1)$$q_{ads}=\frac{\Delta c_{MB}\cdot V_{MB\;aq\;solution}}{m_{sample}}$$
where, Δ*c_MB_*—the difference in the methylene blue concentration before and after adsorption [mg/dm^3^], *V_MB aq solution_*—the volume of the methylene blue aqueous solution used during the adsorption test [dm^3^], and *m_sample_*—is the mass of activated biocarbon used [g].

For the modeling of the equilibrium adsorption isotherms, the Langmuir (Equation (2)) and Freundlich (Equation (3)) equations were used:(2)$$q_{e}=\frac{q_{max}K_{L}c_{e}}{1+K_{L}c_{e}}$$
(3)$$q_{e}=K_{F}c_{e}^{1/n}$$
where, *q_e_*—the equilibrium methylene blue adsorbed amount [mg/g], *q_max_*—the maximal methylene blue adsorbed amount (monolayer capacity) [mg/g], *K_L_*—the Langmuir constant [dm^3^/mg], *c_e_*—the equilibrium concentration of methylene blue [mg/dm^3^], *K_F_*—the Freundlich constant [mg/g (mg/dm^3^)^1/n^] and *n*—the Freundlich parameter (determining the adsorption strength).

The poly(acrylic acid)/PAA amount adsorbed was determined using the static method (the decrease in adsorbate concentration before and after the adsorption process). The polymer solution absorbance was measured using the UV–Vis spectrophotometer Carry 100 (Varian, Palo Alto, CA, USA) and by applying PAA complexation reaction with hyamine 1622. As a result, white-colored compound was formed, which absorbs light at a wavelength of 500 nm [63]. The PAA adsorption was determined after 24 h in the 10–200 ppm polymer concentration range at 25 °C and at pH 6, using 0.01 g of activated carbon. The adsorption kinetics was studied at 25 °C and at pH 6 for the specific time intervals: after 0.17; 0.5; 1; 3; 6; 12 and 24 h. The analogous procedure of polymer concentration determination was used (was 200 ppm). Additionally, the PAA amounts adsorbed (for initial PAA concentration 200 ppm) were determined as a function of solution pH (examined pHs: 2, 4, 6, 8, 10 and 12; at temperature of 25 °C) and temperature (examined temperatures: 25, 30 and 35 °C; pH 6).

Poly(acrylic acid) (Fluka, Saint Louis, MO, USA) was characterized by the weight average molecular weights equal to 2000 Da. PAA as a weak polyelectrolyte with an anionic character (pK_a_ value of about 4.5 [64]) contain carboxyl groups, which undergo dissociation with increasing pH. Thus, at pH values lower than 3, its ionization is negligible; at pH 4.5, it is 0.5 and above pH 6, the polymeric chains are practically totally dissociated [65].

## 4. Conclusions

The conducted studies have shown that waste biomass could be successfully used as a cheap and renewable precursor for the production of activated biocarbons. To a great extent, the physicochemical properties of carbonaceous materials obtained from pistachio nut shells are determined by the variant of the thermochemical conversion applied. Sample activated with orthophosphoric acid has well-developed surface area and porous structure consisting of micropores and small mesopores, whereas analogous product of direct physical activation is characterized by rather poor textural parameters and meso/macroporous structure. Chemically activated biocarbon shows the acidic nature of the surface, whereas for material activated with carbon dioxide, it is alkaline.

Adsorption tests have proven that the obtained carbonaceous materials are characterized by very diverse sorption abilities towards inorganic and organic pollutants. The product of chemical activation of pistachio nut shells turned out to be a much more effective adsorbent in relation to all tested pollutants of different chemical structure and molecular size. Its adsorption capacity toward iodine, methylene blue, a synthetic cationic aromatic dye and poly(acrylic acid), an anionic polymer, reached the level of 1059 mg/g, 183.1 mg/g and 207.9 mg/g, respectively. It has been also proven that the efficiency of the organic impurities removal from the aqueous solutions (in particular polymers) is significantly dependent on the pH and the temperature of the adsorbate–adsorbent system.

## Figures and Tables

**Figure 1 molecules-28-04497-f001:**
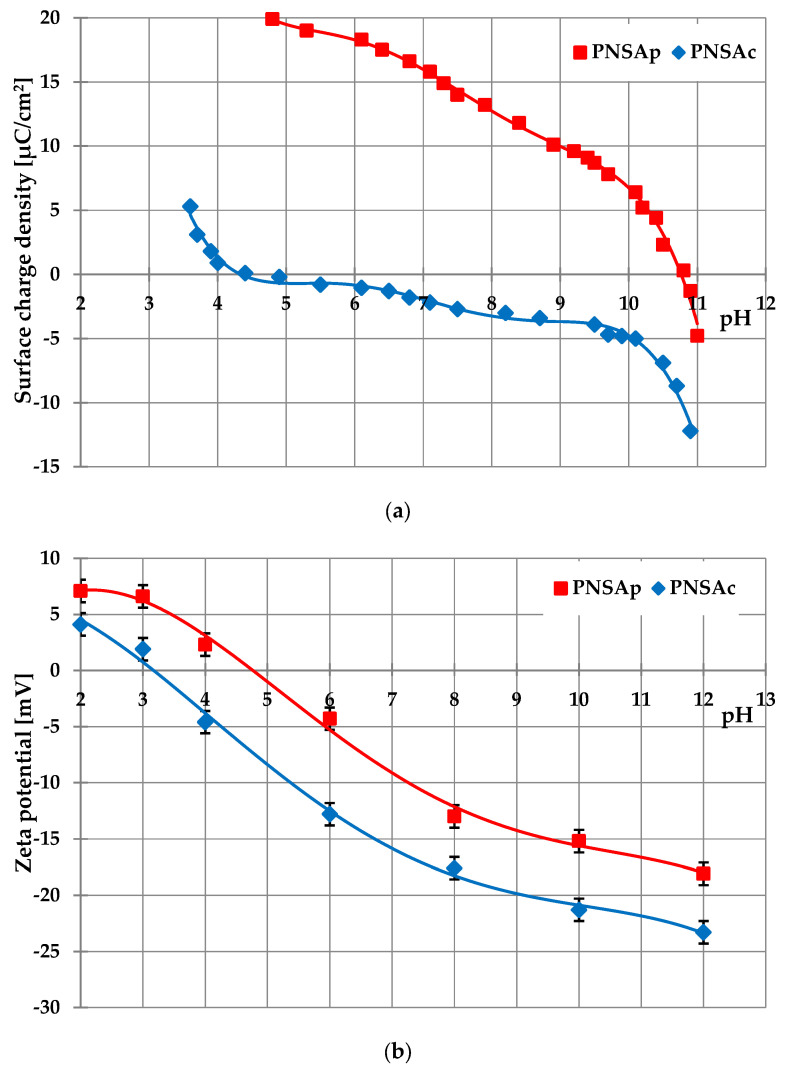
The pH dependencies of the activated carbons’ surface charge density (**a**) and zeta potential (**b**).

**Figure 2 molecules-28-04497-f002:**
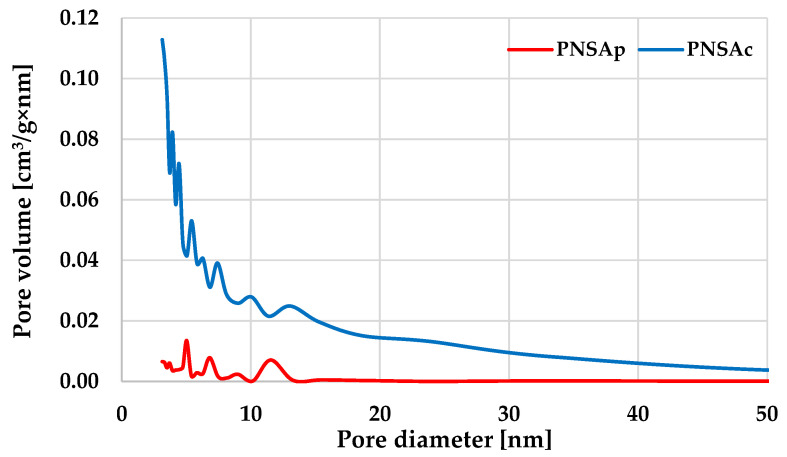
Pore size distribution of the activated biocarbons obtained from pistachio nut shells.

**Figure 3 molecules-28-04497-f003:**
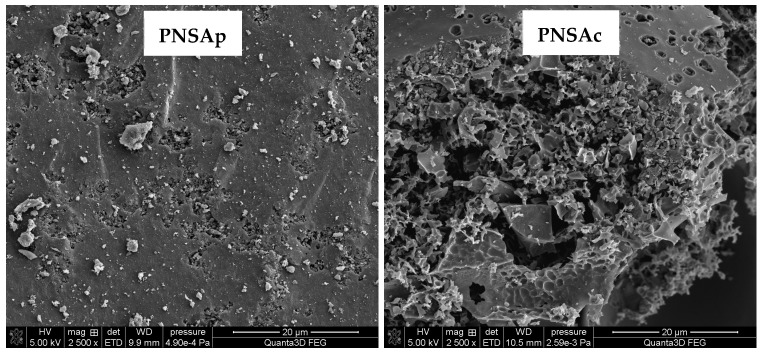
SEM images of the activated biocarbons obtained from pistachio nut shells.

**Figure 4 molecules-28-04497-f004:**
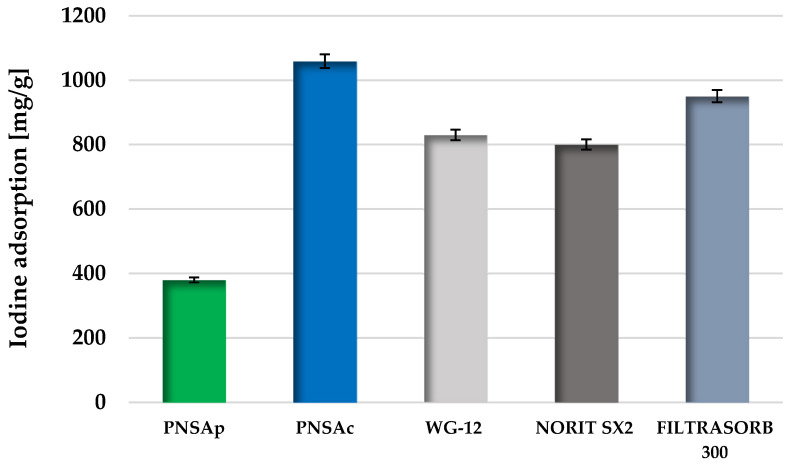
Comparison of iodine adsorption [mg/g] for activated biocarbons prepared from pistachio nut shells and selected commercial products.

**Figure 5 molecules-28-04497-f005:**
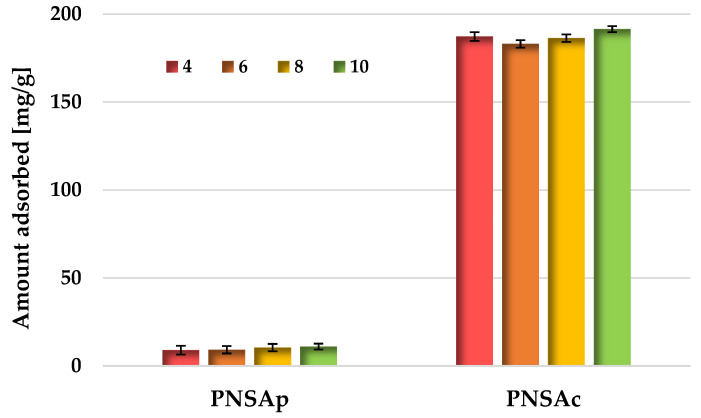
Effect of solution pH on methylene blue adsorption on the activated biocarbons obtained from pistachio nut shells.

**Figure 6 molecules-28-04497-f006:**
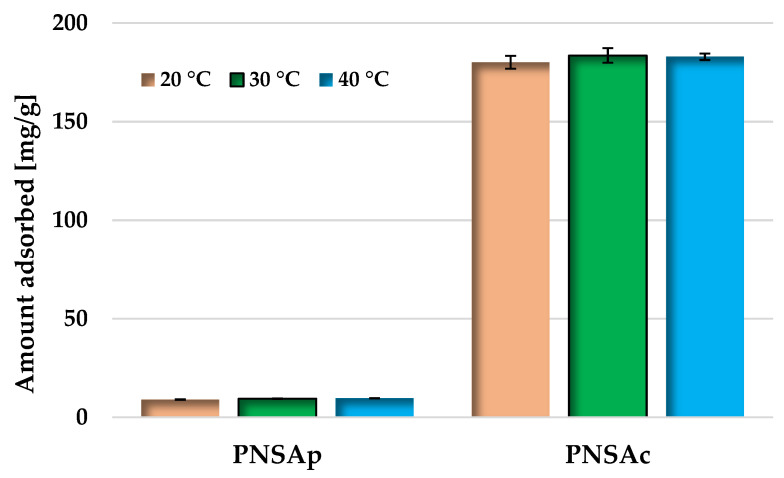
Effect of temperature on methylene blue adsorption on the activated biocarbons obtained from pistachio nut shells.

**Figure 7 molecules-28-04497-f007:**
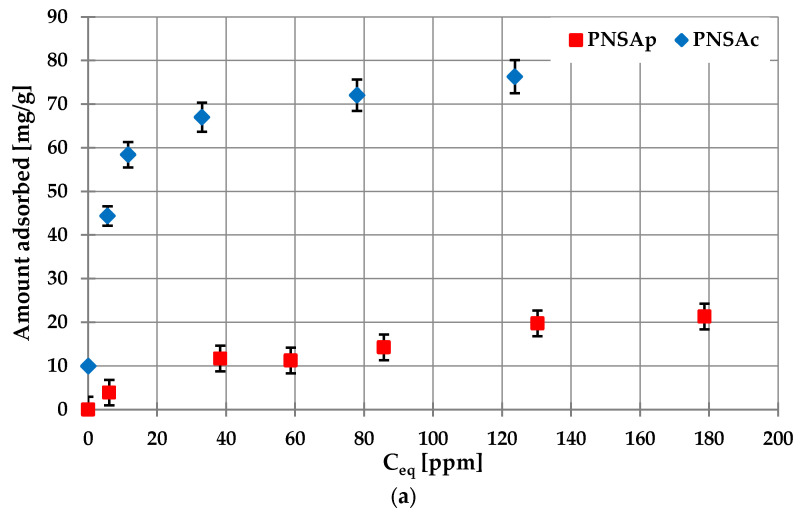
The adsorption isotherms (**a**) and adsorption kinetics (**b**) of poly(acrylic acid) on the activated biocarbons surface obtained at pH 6 and at 25 °C (the initial polymer concentration for kinetic studies was 200 ppm).

**Figure 8 molecules-28-04497-f008:**
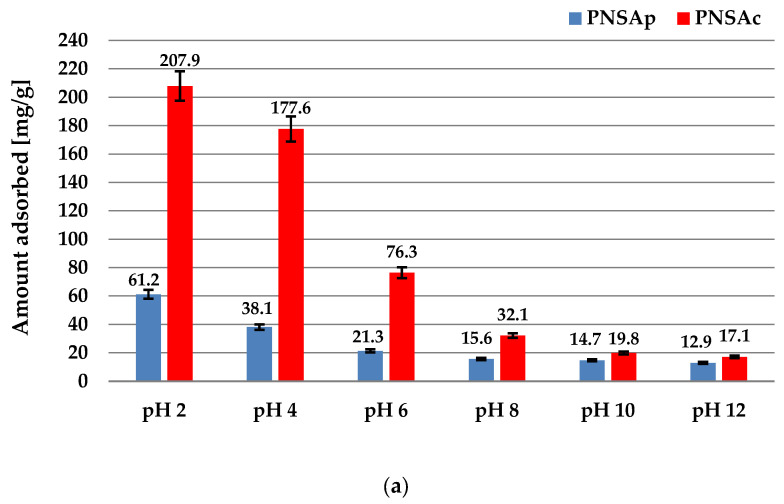
The adsorbed amounts of PAA on the activated carbons surface at different solution pH values at 25 °C (**a**) and temperatures at pH 6 (**b**), the initial polymer concentration was 200 ppm.

**Figure 9 molecules-28-04497-f009:**
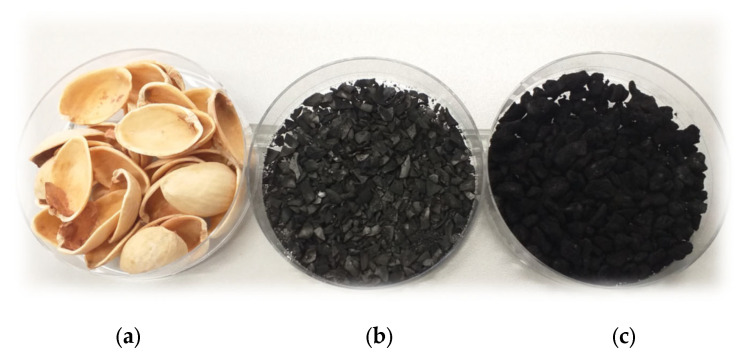
Starting pistachio nut shells (**a**) and products of their physical (**b**) and chemical activation (**c**).

**Table 1 molecules-28-04497-t001:** Elemental composition of the starting pistachio nut shells and activated biocarbons [wt. %].

Sample	Ash	C^daf 1^	H^daf^	N^daf^	S^daf^	O^diff 2^
PNS	0.2	43.9	7.0	0.2	0.1	48.8
PNSAp	1.7	91.7	0.5	0.56	0.1	7.1
PNSAc	3.5	88.6	2.6	0.0	0.1	8.7

^1^ dry-ash-free basis; ^2^ calculated by difference; method error ≤ 0.3%.

**Table 2 molecules-28-04497-t002:** Acidic–basic character of the precursor and the activated biocarbons surface.

Sample	Acidic Groups Content [mmol/g]	Basic Groups Content [mmol/g]	Total Content of Surface Groups [mmol/g]	pH of Aqueous Extracts
PNS	0.65	0.24	0.89	5.29
PNSAp	0.15	0.62	0.77	9.11
PNSAc	0.94	0.07	1.01	3.14

**Table 3 molecules-28-04497-t003:** Textural parameters of the activated biocarbons prepared from pistachio nut shells.

Sample	Total ^1^		Micropore		Micropore Contribution	Mean Pore Size [nm]
	Surface Area [m^2^/g]	Pore Volume [cm^3^/g]	Area [m^2^/g]	Volume [cm^3^/g]		
PNSAp	31	0.059	-	-	-	7.256
PNSAc	1264	1.382	746	0.397	0.287	4.376

^1^ method error in the range from 2 to 5%.

**Table 4 molecules-28-04497-t004:** Langmuir/Freundlich parameters of the isotherms of methylene blue equilibrium adsorption on the activated biocarbons prepared from pistachio nut shells.

Sample	q_exp_	Langmuir Model			Freundlich Model		
		q_max_	K_L_	R^2^	K_F_	1/n	R^2^
PNSAp	9.04	10.41	0.157	0.916	7.179	0.126	0.376
PNSAc	183.11	182.48	0.143	0.999	153.673	0.066	0.910

q_exp_—experimental adsorption capacity [mg/g], q_max_—the maximum adsorption capacity [mg/g], K_L_—the Langmuir adsorption equilibrium constant [dm^3^/mg], K_F_—the Freundlich equilibrium constant [mg/g (mg/dm^3^)^1/n^], 1/n—the intensity of adsorption, R^2^—the correlation coefficients.

**Table 5 molecules-28-04497-t005:** Adsorption capacities towards iodine, methylene blue and poly(acrylic) acid for various adsorbents.

Adsorbent	Maximum Adsorbed Amount [mg/g]	Reference
**Iodine**		
Activated carbon from coconut shell	249	[47]
Activated carbon from acacia wood	381	[48]
Activated carbon from mangosteen peel	1153	[49]
Activated carbon from bean husk	1256	[50]
Activated biocarbon form mugwort	948	[51]
Activated carbon from pistachio nutshells	380 and 1089	This study
**Methylene blue**		
Activated carbon from spent coffee grounds	179	[52]
Activated carbon from bamboo chips	305	[53]
Activated carbon from cashew nut shells	100	[54]
Activated carbon from baobab fruit shell	114	[55]
Activated carbon from pistachio nutshells	9 and 183	This study
**Poly(acrylic acid)**		
Titanium dioxide	24	[56]
Mixed silica-alumina oxide	86	[57]
Activated biocarbon from corncobs	50	[58]
Activated carbon obtained from the nettle herb	273	[59]
Activated carbon from pistachio nutshells	61 and 208	This study

## Data Availability

Data are contained within the article.

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
