# Peer review of "Carbon Adsorbents Obtained from Pistachio Nut Shells Used as Potential Ingredients of Drinking Water Filters"

_molecules, 2023, doi:10.3390/molecules28114497_

Round 1

Reviewer 1 Report

1. The background provides too many general information for the public but more specific background about the the activated carbon and pistachio nut shell should be desired for researchers.

2. The first time the abbreivation name appear, the full name should be given 

3. The discussion of the Fig6 is exaggerated. 

Author Response

First of all, we would like to thank you for your kindly review and suggestions that allowed us to improve the manuscript.

  1. The background provides too many general information for the public but more specific background about the activated carbon and pistachio nut shell should be desired for researchers.

As suggested by the Reviewer, the Introduction has been changed. New references have been added.

  1. The first time the abbreviation name appear, the full name should be given.

Appropriate corrections have been made.

  1. The discussion of the Fig. 6 is exaggerated.

The fragment of the manuscript indicated by the Reviewer has been changed.

Reviewer 2 Report

Paper is fine and can be accepted. English can be slightly polished at proofs reading stage, some misprints are marked in manuscript

The main objective of this manuscript was to produce new activated biocarbons via direct physical and chemical activation of common pistachio nut shells and  to assess their usefulness for the removal of inorganic and organic pollutions  from aqueous solutions. Pretreatment of pistachio nut shells with ortho-314 phosphoric acid was shown by this team to be very efficient  for preparing efficient adsorbents. Regarding the methodology, all is fine and properly addressed, authors are experienced researches in this field and have already published paper in Molecules. Conclusions are sound and based on results, they addressed the main questions posed. All the references, Tables and Figures are fine.

English can be slightly polished at proofs reading stage, some misprints are marked in manuscript

Author Response

First of all, we would like to thank you for your kindly review and suggestions that allowed us to improve the manuscript.

  1. Paper is fine and can be accepted. English can be slightly polished at proofs reading stage, some misprints are marked in manuscript.

English has been checked and corrected.

Reviewer 3 Report

In this manuscript, the results of this research are conveyed thoughtfully and completely, and they are consistent with the experimental findings. However, the authors failed to explain and draw out the novelty of the work, this aspect needs to be improved. This work is worthwhile to be publish in this journal after major revision. The following issues should be addressed:

1. The novelty needs to refinement and should be highlighted in the introduction part.

2. Maybe the author should compare their results clearly with other reported works, highlighting the advantage and disadvantages of their novel composite.

3. The manuscript contains some minor typo/grammar errors, please check all of it.

4. Abstract not targeted; the authors should rephrase it.

5. Error bars need to be displayed for all experimental data in the graphs

6. some references should be added in the preparation methods of activated biocarbon

https://doi.org/10.21608/DUSJ.2023.291058

https://doi.org/10.3390/ma16062170

Hence, I recommend it accepted for publication after Major revisions.

Author Response

First of all, we would like to thank you for your kindly review and valuable suggestions that allowed us to improve significantly our manuscript.

  1. The novelty needs to refinement and should be highlighted in the introduction part.

According to Reviewer suggestion, the novelty of manuscript has been underlined.

  1. Maybe the author should compare their results clearly with other reported works, highlighting the advantage and disadvantages of their novel composite.

The revised version of the manuscript includes a comparison of the obtained results with the literature data

  1. The manuscript contains some minor typo/grammar errors, please check all of it.

The whole manuscript has been checked and corrected.

  1. Abstract not targeted; the authors should rephrase it.

Abstract has been improved according to Reviewer advice.

  1. Error bars need to be displayed for all experimental data in the graphs.

According to Reviewer sugesstion error bars have been added to Figures.

  1. Some references should be added in the preparation methods of activated biocarbon.

https://doi.org/10.21608/DUSJ.2023.291058; https://doi.org/10.3390/ma16062170.

A new References has been added and the Introduction section has been modified.

Round 2

Reviewer 3 Report

Accepted in the present form